# Safety, Immunogenicity, and Efficacy of COVID-19 Vaccines in Children and Adolescents: A Systematic Review

**DOI:** 10.3390/vaccines9101102

**Published:** 2021-09-29

**Authors:** Meng Lv, Xufei Luo, Quan Shen, Ruobing Lei, Xiao Liu, Enmei Liu, Qiu Li, Yaolong Chen

**Affiliations:** 1Department of Nephrology, Children’s Hospital of Chongqing Medical University, Chongqing 400014, China; lvm2016@163.com; 2National Clinical Research Center for Child Health and Disorders, Ministry of Education Key Laboratory of Child Development and Disorders, China International Science and Technology Cooperation Base of Child Development and Critical Disorders, Children’s Hospital of Chongqing Medical University, Chongqing 400014, China; shenquan@whu.edu.cn (Q.S.); leiruobing1009@163.com (R.L.); emliu186@126.com (E.L.); 3Chongqing Key Laboratory of Pediatrics, Chongqing 400014, China; 4School of Public Health, Lanzhou University, Lanzhou 730000, China; luoxf2016@163.com (X.L.); liuxiao20@lzu.edu.cn (X.L.); 5Chevidence Lab Child & Adolescent Health, Department of Pediatric Research Institute, Children’s Hospital of Chongqing Medical University, Chongqing 400014, China; 6Department of Respiratory Medicine, Children’s Hospital of Chongqing Medical University, Chongqing 400014, China; 7Institute of Health Data Science, Lanzhou University, Lanzhou 730000, China; 8Evidence-Based Medicine Center, School of Basic Medical Sciences, Lanzhou University, Lanzhou 730000, China; 9WHO Collaborating Centre for Guideline Implementation and Knowledge Translation, Lanzhou 730000, China; 10Guideline International Network Asia, Lanzhou 730000, China; 11Key Laboratory of Evidence Based Medicine and Knowledge Translation of Gansu Province, Lanzhou University, Lanzhou 730000, China; 12Lanzhou University GRADE Center, Lanzhou 730000, China

**Keywords:** COVID-19, vaccine, children, adolescents, systematic review

## Abstract

Aim: To identify the safety, immunogenicity, and protective efficacy of COVID-19 vaccines in children and adolescents. Methods: We conducted a systematic review of published studies and ongoing clinical studies related to the safety, immunogenicity, and efficacy of COVID-19 vaccine in children or adolescents (aged < 18 years). Databases including PubMed, Web of Science, WHO COVID-19 database, and China National Knowledge Infrastructure (CNKI) were searched on 23 July 2021. International Clinical Trials Registry Platform (ICTRP) was also searched to identify ongoing studies. Results: Eight published studies with a total of 2852 children and adolescents and 28 ongoing clinical studies were included. Of the eight published studies, two were RCTs, two case series, and four case reports. The investigated COVID-19 vaccines had good safety profiles in children and adolescents. Injection site pain, fatigue, headache, and chest pain were the most common adverse events. A limited number of cases of myocarditis and pericarditis were reported. The RCTs showed that the immune response to BNT162b2 in adolescents aged 12–15 years was non-inferior to that in young people aged 16–25 years, while with 3 μg CoronaVac injection the immune response was stronger than with 1.5 μg. The efficacy of BNT162b2 was 100% (95% CI: 75.3 to 100), based on one RCT. Of the 28 ongoing clinical studies, twenty-three were interventional studies. The interventional studies were being conducted in fifteen countries, among them, China (10, 43.5%) and United States(9, 39.1%) had the highest number of ongoing trials. BNT162b2 was the most commonly studied vaccine in the ongoing trials. Conclusion: Two COVID-19 vaccines have potential protective effects in children and adolescents, but awareness is needed to monitor possible adverse effects after injection. Clinical studies of the COVID-19 vaccination in children and adolescents with longer follow-up time, larger sample size, and a greater variety of vaccines are still urgently needed.

## 1. Background

One and a half year have passed since the beginning of the coronavirus disease 2019 (COVID-19) pandemic. Yet the epidemic is still not under control. With over 200 million confirmed severe acute respiratory syndrome coronavirus 2 (SARS-CoV-2) infections and over 4 million COVID-19 related deaths, COVID-19 has brought great suffering and devastation to people worldwide.

Vaccines, as an effective way to prevent and control disease infections, stimulate the human immune system to produce antibodies, thus increasing immunity to the disease and generating protection for the immunized individual [1]. Vaccination aims to curb the spread of the disease and helps to potentially achieve herd immunity. As of 18 September 2021, twenty-two COVID-19 vaccines worldwide have been approved [1]. However, we have little knowledge of the efficacy and safety of COVID-19 vaccines in children and adolescents. Given that children and adolescents account for approximately one quarter of the world’s population [2], promoting vaccination of children and adolescents is also crucial to end the spread of COVID-19.

The development of COVID-19 vaccine has been in full swing since the COVID-19 outbreak. Studies have shown that the current COVID-19 vaccines are effective and safe in adults [3,4,5,6]. Several international organizations and countries have also developed guidelines for different aspects of COVID-19 vaccination, including vaccination of special populations, management of adverse reactions, and cautions for vaccination [7,8,9]. However, the efficacy of protection and adverse effects of COVID-19 vaccines in children and adolescents remains unclear despite a large number of clinical trials being conducted. Furthermore, children and adolescents have less severe COVID-19 symptoms than adults [10], and they likely play a limited role in spreading the infection to others. Therefore, more high-quality clinical studies are still needed to determine whether COVID-19 vaccination should be recommended for children at the moment [11]. In addition, children are a population group with special needs and features, and the attitude of parents or guardians toward the COVID-19 vaccine is also an essential factor affecting children’s vaccination. To explore and promote COVID-19 vaccination in children and adolescents, The National Clinical Research Center for Child Health and Disorders (Chongqing, China) initiated an international guideline for the management of COVID-19 in children and adolescents [12] that also contains the question of whether and how children and adolescents should be vaccinated against COVID-19. To answer this question, we conducted a systematic review to estimate the safety, immunogenicity, and protective efficacy of the COVID-19 vaccine in children and adolescents, covering both completed and ongoing studies and trials.

## 2. Methods

We conducted this systematic review in accordance with the Preferred Reporting Items for Systematic Reviews and Meta-Analysis (PRISMA) (see Appendix A for PRISMA checklist) [13] and the Cochrane Handbook for Systematic Reviews of Interventions [14]. We have registered this systematic review at OSF REGISTRIES (DOI:10.17605/OSF.IO/JC32H, accessed on 3 August 2021).

### 2.1. Inclusion and Exclusion Criteria

We included published studies and ongoing clinical studies related to the safety, immunogenicity, and efficacy of COVID-19 vaccine in children or adolescents (aged < 18 years). The study design was limited to primary studies, including randomized clinical trials (RCTs), non-randomized trials, and observational studies. We also included ongoing studies registered at the International Clinical Trials Registry Platform (ICTRP).

We excluded articles from which we could not extract data specifically on children or adolescents or if we could not access the full text, conference proceedings, and study protocols. For ongoing studies, we only included registration records if the aim of the study was to determine the safety, immunogenicity, or efficacy of COVID-19 vaccine in children and adolescents.

### 2.2. Search Strategy

We systematically searched Medline (via PubMed), Web of Science, World Health Organization (WHO) COVID-19 database, and China National Knowledge Infrastructure (CNKI), from their inception to 23 July 2021 to identify studies that met our eligibility criteria. The search strategy combined terms from three themes: (1) COVID-19, (2) vaccine, and (3) children and adolescents (see detailed search strategy in Appendix A). All search strategies were developed and retrieved independently by two investigators (ML and XL) and then cross-checked. We first developed a search strategy for Medline, and after reaching agreement adapted this strategy for other databases. In addition to the literature databases, we searched ICTRP to identify ongoing studies. We also searched Google Scholar and reference lists of identified articles to avoid missing potentially relevant literature.

### 2.3. Literature Screening

The screening process included three phases. First, one investigator removed duplicates from the retrieved records. Following this, four investigators (ML, XL, RL, and QS) screened all identified records independently by reading titles and abstracts. If the information in the title and abstract was insufficient, the full text was obtained for review. Disagreements were solved by consensus with the senior researcher (YC). We used Endnote 20.0.1 software in the entire screening process.

### 2.4. Data Extraction

The following data were extracted from the completed studies: (1) basic information: publication date, country, study design, name of the vaccine; (2) information of the participants: age, sample size, sex distribution; and (3) outcome information: safety, immunogenicity, and efficacy of COVID-19. For the ongoing clinical studies, we extracted the registration date, country, recruitment status, participants’ age, target sample size, intervention, and primary outcome. All data were independently extracted by two investigators (ML and XL) using a predesigned extraction sheet.

### 2.5. Risk of Bias Assessment

Two investigators (ML and XL) assessed the methodological quality of the original studies to ensure the reliability of the findings. We used the Risk of Bias tool recommended by Cochrane Collaboration [15] to assess randomized trials. The tool consists of six domains of bias (selection bias, performance bias, detection bias, attrition bias, reporting bias, and other bias). For case-control and cohort studies we used the Newcastle-Ottawa Scale (NOS) [16].; for case series and case reports the checklist proposed by Murad et al. [17]; and for cross-sectional studies the checklist of the Joanna Briggs Institute (JBI) [18].

### 2.6. Data Analysis

We descriptively presented the main findings on safety, immunogenicity, and efficacy of COVID-19 vaccine in children or adolescents. Microsoft Excel 16.51 (2019) was used for data processing and analysis. We considered to conduct a quantitative meta-analysis if at least two studies were included and the heterogeneity between the studies in terms of outcomes, population characteristics, and type of vaccine was low (I^2^ ≤ 50%). For ongoing clinical studies, we also presented the numbers of trials by country and type of vaccine. Adobe Illustrator was used to visually present the number of ongoing clinical trials of COVID-19 vaccine in children or adolescents worldwide.

## 3. Results

### 3.1. Literature Search

Our initial search revealed 3092 records, of which 931 were excluded as duplicates. After screening the titles and, if necessary, full texts, eight published studies [19,20,21,22,23,24,25,26] with 2852 children or adolescents and 28 ongoing clinical studies targeting to recruit a total of 122,442 participants were included. The study selection process is shown in detail in Figure 1.

### 3.2. Characteristics of the Included Clinical Studies

Among the eight published studies included, two were RCTs [19,20], two were case series [21,22], and four were case reports [23,24,25,26]. Five studies were conducted in the United States, and one in China, France, and Israel each. The studies were restricted to adolescents with the exception of one RCT that included children aged between 3 and 17 years. In one study the participants received CoronaVac COVID-19 vaccine developed by Sinovac Life Sciences, and in the other seven the participants received BNT162b2 mRNA COVID-19 vaccine developed by Pfizer-BioNTech. The characteristics of the included studies are summarized in Table 1.

### 3.3. Quality of Included Studies

The overall methodological quality of the two included RCTs was high and the risk of bias low (Table 2). In the rest of the studies (case series and case reports), we did not assess two of the eight items of the Murad et al. [17] checklist, “Was there a challenge/rechallenge phenomenon” and “Was there a dose-response effect?”, because they were not applicable. One study complied with five of the remaining six items, three with four items, one with three items, and one with two items. The method of case selection was unclear in all included case series and case reports. Only two case reports or case series reported the item “were other alternative causes that may explain the observation ruled out?”, and in three studies the follow-up time was not long enough for outcomes to occur.

### 3.4. Safety of COVID-19 Vaccines

The most common adverse event in the two RCTs was injection site pain [20,21]. Besides that, fever, headache, and fatigue were also frequently reported. Most adverse events were not severe. No deaths were reported. A case series [22] that included 13 patients with solid tumor also showed that mild-to-moderate injection site pain was the most frequent adverse event (6 patients).

Besides, a few diagnosed myocarditis and/or pericarditis cases related to COVID-19 vaccine were reported in some studies. All cases occurred following the second dose of BNT162b mRNA COVID-19 vaccination. We summarized the basic information of 27 cases from included studies (Table 3). The median age was 16 years (range, 12–17 years). Most patients were male (26, 96.3%). Median time of onset was 3 days after receiving the vaccine (range, 1–4 days). All patients had chest pain.

### 3.5. Immunogenicity of the COVID-19 Vaccines

The two included RCTs indicated that the investigated COVID-19 vaccines, CoronaVac and BNT162b2, were immunogenic in children and adolescents. Frenck et al. [20] reported that the immune response to BNT162b2 in 12–15 year old adolescents was noninferior to that in young adults aged 16–25 (geometric mean ratio (GMR) = 1.75, 95% CI: 1.47~2.10), indicating even a better response in 12–15 years group than in young adults. Han et al. [19] found that in Phase 1, the seroconversion of neutralizing antibody after the second dose was 100% both in 1.5 μg group and 3.0 μg group with geometric mean titer (GMT) of 55.0 (95% CI 38.9–77.9) and 117.4 (87.8–157.0), respectively (*p* = 0.0012). In Phase 2, the seroconversion rates were 96·8% (95% CI: 93.1–98.8) and 100% (95% CI: 98.0–100.0) in the 1.5 μg group and the 3.0 μg group, respectively (*p* = 0.030).

### 3.6. Efficacy of the COVID-19 Vaccines

The RCTs on BNY162b2 [20] showed that the efficacy of the vaccine in children and adolescents was 100% (95% CI: 75.3~100). The other RCT on CoronaVac did not assess vaccine efficacy.

### 3.7. Ongoing Clinical Studies

We identified 28 ongoing clinical studies with a total target sample size of 122,442 (see Appendix A for ongoing clinical trials on COVID-19 vaccination in children and adolescents). Twenty-three were interventional studies (including one Phase 1 trial; six Phase1/2 trials; six Phase 2 trials; four Phase 2/3 trials; three Phase 3 trials; one Phase 4 trial; and one where the phase was not clear) and five were observational studies. The minimum age of eligible participants was 6 months. Twenty-seven studies reported the name of vaccine they planned to use and there were a total of 15 different vaccine candidates of the following five major types: mRNA (13 studies), inactivated (7 studies), protein subunit (four studies), non-replicating viral vector (four studies), and replicating viral vector (one studies).

The interventional clinical trials were being conducted in 15 countries, the highest numbers of planned trials being in China (10 trials, 43.5%) and the United States (9 trials, 39.1%). BNT162b2 was the most common vaccine (6 trials, 26.1%). Figure 2 shows the countries with ongoing clinical trials and vaccines used in trials.

## 4. Discussion

### 4.1. Principal Findings

Our review identified eight completed studies and 28 ongoing clinical studies of COVID-19 vaccines in children and adolescents. The investigated COVID-19 vaccines had good safety profiles, most adverse effects were mild or moderate, such as injection site pain, fatigue, headache, and chest pain. Some studies reported a few cases of myocarditis and pericarditis. The immune response to the BNT162b2 vaccine in adolescents aged 12–15 years was non-inferior to that in young people aged 16–25 years, and CoronaVac injection had a stronger immune response with a 3.0 μg than 1.5 μg dose. According to the one RCT on BNT162b2, no cases of COVID-19 in adolescents aged 12–15 years were detected. Clinical trials on children and adolescents are being conducted all over the world with a large number of different vaccines.

Children and adolescents, as a special population, present many influencing factors to consider when administering vaccines. Vaccine efficacy and safety are the most important considerations for children and their parents [27]. It is therefore important to demonstrate that vaccines are safe and protective before they are administered to children and adolescents. During an average influenza season, approximately 9.8% of children aged 0–14 year present with influenza [28]. After vaccination against influenza A (H1N1), 90.3% of children and adolescents aged 10–17 years developed protective antibodies, and no serious adverse reactions were seen [29,30]. Similarly, when the COVID-19 outbreak emerged, researchers actively promoted the development of vaccines with the expectation that vaccination could protect healthy population. Our study showed that two vaccines have shown to be effective and safe in pediatric populations. However, the evidence for both vaccines was based on single RCTs, and these two studies both had limitations such as the small sample size and lack of long-term data on safety and immunogenicity data. In particular, the risk of myocarditis and pericarditis should be closely monitored. Most cases of myocarditis and pericarditis associated with the COVID-19 vaccine were mild, and mostly affected children were male. Schauer et al. [26] estimated an incidence of myopericarditis of 0.008% in adolescents 16–17 years of age and 0.01% in those aged 12 through 15 years following the second dose.

Another important factor to consider for vaccination of children and adolescents is the risk of multisystemic inflammatory syndrome in children (MIS-C). In April 2020, children infected with SARS-CoV-2 presenting symptoms similar to incomplete Kawasaki disease (KD) or toxic shock syndrome were documented in the UK [31]. Since then, children with similar symptoms have been reported in other parts of the world as well [32,33,34]. This condition was subsequently named as MIS-C. The overall mortality of MIS-C is approximately 1–2% [35]. The decision to vaccinate should be made by weighing the risk of exposure, reinfection, and severe disease following infection against the uncertain safety of vaccination in such individuals. Whereas no directly relevant studies have confirmed the association of MIS-C with COVID-19 vaccination, a systematic review published in 2017 [36] identified 27 observational studies and case reports of KD. These showed that diphtheria-tetanus-pertussis (DTP)-containing vaccines, Haemophilus influenzae type b (Hib) conjugate vaccine, influenza vaccine, hepatitis B vaccine, 4-component meningococcal serogroup B (4CMenB) vaccine, measles-mumps-rubella (MMR)/MMR-varicella vaccines, pneumococcal conjugate vaccine (PCV), rotavirus vaccine (RV), yellow fever vaccine, and Japanese encephalitis vaccine did not increase the risk of KD. Thus, children and adolescents at high risk of severe COVID-19 or those with specific comorbidities should be considered to be prioritized in vaccination. More research is needed to clarify to what extent COVID-19 vaccines can mitigate the risks and bring benefits.

To date, 22 COVID-19 vaccines have been approved throughout the world, more than 1/3 of which are inactivated, and 138 vaccines are under development and exploitation. More than 300 clinical trials of COVID-19 vaccines have been registered or published [37,38]. Studies have shown that most COVID-19 vaccines are safe and effective in adults aged ≥ 18 years. Overall, in phase 2 and 3 RCTs, mRNA- and adenoviral vector-based COVID-19 vaccines had 94.6% (95% CI 0.936–0.954) and 80.2% (95% CI 0.56–0.93) efficacy, respectively [3,4,5], with good acceptability [6] and safety [39]. Only two RCTs on children and adolescents have been published in peer-reviewed journals so far, both of which found that the respective vaccines, BNT162b2 and CoronaVac, are safe and effective. Institutions including WHO, Centers for Disease Control and Prevention (CDC), the Food and Drug Administration (FDA), the Canadian Pediatric Society have already authorized emergency use of BNT162b2 in children and adolescents aged 12 years and above [40,41,42,43]. European Medicines Agency (EMA) has also approved the Spikevax (previously COVID-19 Vaccine Moderna) vaccine for adolescents aged 12 to 17 years, based on the evidence from an ongoing study [44]. Although these guidelines gave recommendations on vaccinating children or adolescents from the perspective of Western countries, we still need to wait for more evidence from more countries and regions to better understand how COVID-19 vaccines work in different populations. With the more than twenty ongoing clinical trials, their findings may continue to offer clues of better protecting younger generations from COVID-19.

Public health authorities in countries that have approved COVID-19 vaccine in children and adolescents should also consider multiple aspects in their decision-making. European Centre for Disease Prevention and Control issued a set of eight interim considerations from the view of the overall potential public health impact of COVID-19 vaccination of adolescents [45]. Opel et al. suggested nine criteria to consider when evaluating antigens for inclusion in mandatory school immunization programs, which were categorized into vaccine-related, disease-related, and implementation-related [11]. We currently know however too little about the performance of COVID-19 vaccines or the epidemiology of SARS-CoV-2 in children to make any definitive judgment about whether COVID-19 vaccine should be mandatory in children, especially those under 12. Authorities should closely monitor and continually assess the benefits and potential risks of vaccination in children and adolescents. In addition, the acceptability of the COVID-19 vaccine among both the children themselves as well as their parents and guardians is a major influencing factor on the likelihood of children getting vaccinated. Studies have shown that approximately 80% of parents were reluctant to enroll their children in clinical studies of the COVID-19 vaccine [46] and approximately half of Chinese parents showed hesitancy on taking the COVID-19 vaccine for their children [47]. Therefore, it is necessary to educate parents and children about the vaccine to increase vaccination rates while ensuring the efficacy and safety of vaccines [48]. Furthermore, factors such as national policy, religion, culture, and other routine immunization procedures need to be taken into account in the administration of COVID-19 vaccine to children.

### 4.2. Potential Impact for Future Research and Practice

Our study included only two RCTs on COVID-19 vaccination in children and adolescents, one investigating CoronaVac developed by Sinovac and one BNT162b2 developed by Pfizer/BioNTech. For the vast majority of vaccines clinical studies are either ongoing but not completed, or not yet planned. For future research, we recommend paying attention to the following three aspects. First, more clinical studies on the protective efficacy and safety of COVID-19 vaccine in children and adolescents need to be conducted. Second, systematic reviews of factors affecting COVID-19 vaccination in children and adolescents, willingness to be vaccinated, and methods to promote vaccination, are needed. This includes also updating this systematic review when more studies, in particular RCTS, on COVID-19 in children and adolescents become available. Third, evidence-based guidelines for COVID-19 vaccination in children and adolescents are needed to promote and standardize vaccination in children and adolescents. Policymakers should develop policies for COVID-19 vaccination in children and adolescents based on the best current evidence in the future, and parents and guardians should be guided by policies that actively encourage and support their children to be vaccinated against COVID-19.

### 4.3. Strengths and Limitations

This paper is, to the best of our knowledge, the first systematic review on the safety, immunogenicity, and protective efficacy of COVID-19 vaccination in children and adolescents. We systematically searched key databases and websites to conduct a comprehensive evaluation and analysis of published studies and registry data records. However, this paper also has some limitations. First, we did not conduct a meta-analysis in this study, because of the heterogeneity in participant characteristics, outcomes, and study designs. Second, this study only included articles published in English. However, as the amount of evidence published so far is known to be limited, it is reasonable to expect that the studies we included covered most of the knowledge up to now. Finally, some studies that included children and adolescents did not report the age and outcome among these age groups separately. Given the limited time, we excluded these studies instead of contacting authors to request access to original data.

## 5. Conclusions

Our review found high rates of immunogenicity and vaccine efficacy in children and adolescents. This is a clear indicators that the vaccines are effective, and the RCTs also did not find any major issues with safety. Nevertheless, awareness is needed to monitor the possible adverse effects. Although most adverse events observed in the trials were mild, we identified a limited number of cases of myocarditis and pericarditis among the vaccinated children and adolescents, from several different studies. This shows also that particularly in the current situation where RCTs are still limited, it is important to include all existing evidence, also from individual case reports, in systematic reviews. Real-world data can also reveal findings that may not be observed in the well-controlled RCT settings. It is crucial that more clinical studies with sufficiently long follow-up time, large sample size, and using different types of vaccine are conducted in the future. Evidence-based guidelines are urgently needed to inform policymakers, children and adolescents, and their parents and guardians about the benefits and risks of vaccination against COVID-19.

## Figures and Tables

**Figure 1 vaccines-09-01102-f001:**
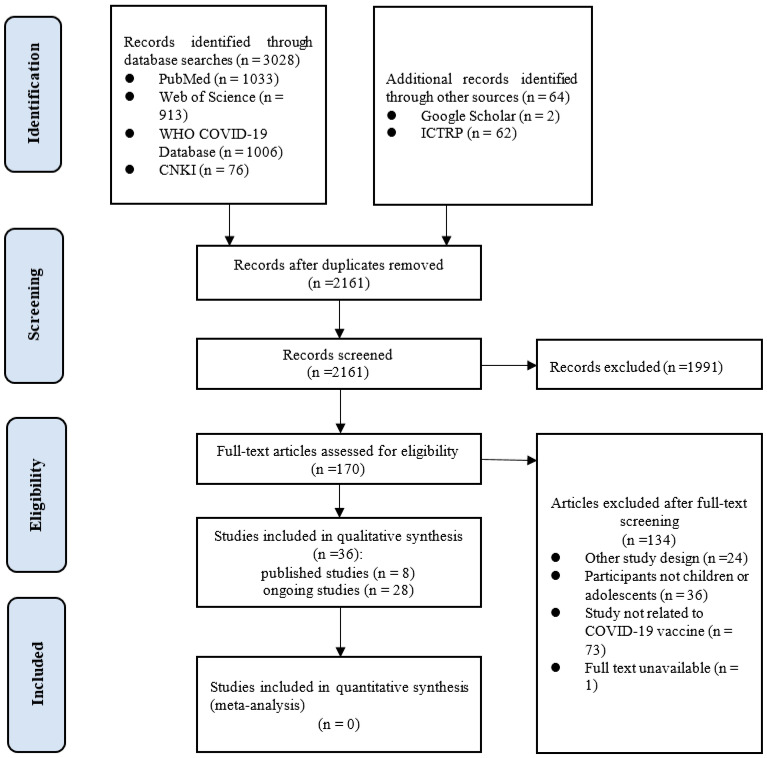
Study selection process (WHO: World Health Organization; COVID-19: coronavirus disease 2019; CNKI: China National Knowledge Infrastructure; ICTRP: International Clinical Trials Registry Platform).

**Figure 2 vaccines-09-01102-f002:**
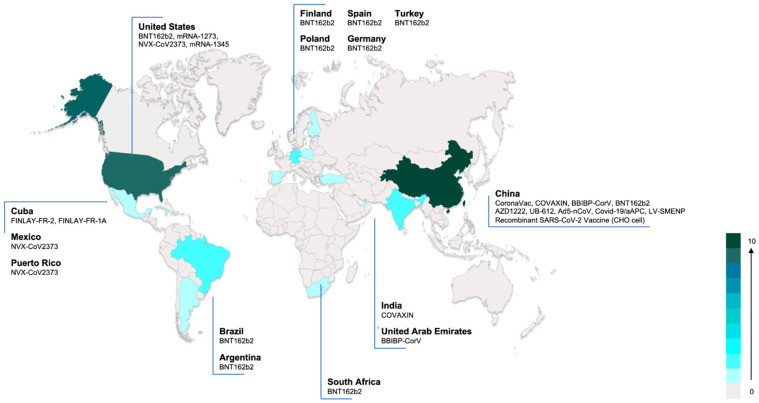
Ongoing interventional COVID-19 vaccine trials in children and adolescents worldwide. Color in the figure indicates the number of ongoing vaccine trials in each country.

**Table 1 vaccines-09-01102-t001:** Basic characteristics of included clinical studies (n = 8).

Name of Vaccine	Participants	Sample Size	Follow-Up Duration	Study Design	Country	Funding	Reference
CoronaVac	Healthy children and adolescents aged 3–17 years	552	4.1 months	RCT Phase 1–2	China	Public/nonprofit (Chinese National Key Research and Development Program and Beijing Science and Technology Program)	Han et al., 2021 [19]
BNT162b2	Adolescents aged 12–15 years with no previous COVID-19 diagnosis or SARS-CoV-2 infection	2264	4.7 months	RCT Phase 3	USA	Private (BioNTech and Pfizer)	Frenck et al., 2021 [20]
BNT162b2	Adolescents and young adults aged 16 years with solid tumor older than	9	NR *	Case series	France	NR *	Riviere et al., 2021 [21]
BNT162b2	Adolescents aged 16–18 years	7	NR *	Case series	Israel	None	Snapiri et al., 2021 [22]
BNT162b2	An adolescent aged 17 years	1	2 weeks	Case report	USA	NR *	Minocha et al., 2021 [23]
BNT162b2	A previously healthy adolescent aged 16 years	1	2 weeks	Case report	USA	NR *	McLean et al., 2021 [24]
BNT162b2	Healthy adolescents 14–18 years	5	unclear	Case report	USA	None	Marshall et al., 2021 [25]
BNT162b2	Children and adolescents aged 12–17 years	13	3 months	Case report	USA	NR *	Schauer et al., 2021 [26]

* NR: not reported.

**Table 2 vaccines-09-01102-t002:** Quality assessment of included studies.

**Risk of Bias in the Included Rcts Assessed by the Risk of Bias Tool**								
Selection bias		Performance bias	Detection bias	Attrition bias	Reporting bias		Other bias	Study
Random sequence generation	Allocation concealment	Blinding of participants and personnel	Blinding of outcome assessment	Incomplete outcome data	Selective reporting		Anything else, ideally prespecified	
low	low	low	low	low	low		low	Han et al., 2021 [19]
low	low	low	low	unclear	low		low	Frenck et al., 2021 [20]
**Methdological quality in the case series and case reports assessed by Murad et al. checklist**								
Selection	Ascertainment		Causality				Reporting	Study
Does the patient(s) represent(s) the whole experience of the investigator (centre) or is the selection method unclear to the extent that other patients with similar presentation may not have been reported?	Was the exposure adequately ascertained?	Was the outcome adequately ascertained?	Were other alternative causes that may explain the observation ruled out?	Was there a challenge/rechallenge phenomenon?	Was there a dose-response effect?	Was follow-up long enough for outcomes to occur?	Is the case(s) described with sufficient details to allow other investigators to replicate the research or to allow practitioners make inferences related to their own practice?	
0	1	1	0	N/A	N/A	0	0	Revon-Riviere et al., 2021 [21]
0	1	1	0	N/A	N/A	0	1	Snapiri et al., 2021 [22]
0	1	1	0	N/A	N/A	1	1	Minocha et al., 2021 [23]
0	1	1	0	N/A	N/A	1	1	McLean et al., 2021 [24]
0	1	1	1	N/A	N/A	0	1	Marshall et al., 2021 [25]
0	1	1	1	N/A	N/A	1	1	Schauer et al., 2021 [26]

0 = no; 1 = yes; N/A: Not applicable.

**Table 3 vaccines-09-01102-t003:** Basic information of diagnosed myocarditis and/or pericarditis cases (n = 27).

Vaccination	Age	Sex	Symptoms	Diagnosis	Time of Onset (Days Since Vaccination)	Length of Hospitalization (Days)	Study
BNT162b2, second dose	17	M	Chest pain	Perimyocarditis	3	4	Snapiri et al., 2021 [22]
BNT162b2, second dose	16	M	Chest pain	Perimyocarditis	1	6	Snapiri et al., 2021 [22]
BNT162b2, second dose	16	M	Chest pain, cough	Perimyocarditis	2	6	Snapiri et al., 2021 [22]
BNT162b2, second dose	16	M	Chest pain, nausea	Perimyocarditis	3	4	Snapiri et al., 2021 [22]
BNT162b2, second dose	17	M	Chest pain, headache	Perimyocarditis	1	5	Snapiri et al., 2021 [22]
BNT162b2, second dose	16	M	Chest pain, dyspnea, diarrhea, fever	Perimyocarditis	2	5	Snapiri et al., 2021 [22]
BNT162b2, second dose	17	M	Chest pain, dyspnea	Perimyocarditis	3	3	Snapiri et al., 2021 [22]
BNT162b2, second dose	17	M	Chest pain, fever, body aches,	Myocarditis	1	6	Minocha et al., 2021 [23]
BNT162b2, second dose	16	M	Chest pain	Myopericarditis	2.5	6	McLean et al., 2021 [24]
BNT162b2, second dose	16	M	Chest pain, bilateral arm pain, fever, fatigue, nausea, vomiting, anorexia, headache	Myocarditis	2	6	Marshall et al., 2021 [25]
BNT162b2, second dose	17	M	Chest pain, bilateral arm pain, numbness, paresthesia	Myopericarditis	2	2	Marshall et al., 2021 [25]
BNT162b2, second dose	17	M	Chest pain, bilateral arm pain, abdominal pain, fever, nausea, vomiting, anorexia, SOB, palpitations	Myocarditis	4	5	Marshall et al., 2021 [25]
BNT162b2, second dose	16	M	Chest pain, SOB	Myocarditis	3	3	Marshall et al., 2021 [25]
BNT162b2, second dose	14	M	Chest pain, fever, SOB	Myopericarditis	2	4	Marshall et al., 2021 [25]
BNT162b2, second dose	16	M	Chest pain, fever, chills, myalgias, headache, SOB	Myopericarditis	2	1	Schauer et al., 2021 [26]
BNT162b2, second dose	16	M	Chest pain, fever, myalgias	Myopericarditis	2	1	Schauer et al., 2021 [26]
BNT162b2, second dose	16	M	Chest pain, myalgias, headache	Myopericarditis	3	3	Schauer et al., 2021 [26]
BNT162b2, second dose	17	M	Chest pain, fever, malaise	Myopericarditis	3	1	Schauer et al., 2021 [26]
BNT162b2, second dose	15	M	Chest pain, myalgias, SOB	Myopericarditis	2	2	Schauer et al., 2021 [26]
BNT162b2, second dose	15	F	Chest pain, vomiting	Myopericarditis	3	1	Schauer et al., 2021 [26]
BNT162b2, second dose	15	M	Chest pain, fevers, SOB	Myopericarditis	3	3	Schauer et al., 2021 [26]
BNT162b2, second dose	15	M	Chest pain, chills	Myopericarditis	3	3	Schauer et al., 2021 [26]
BNT162b2, second dose	12	M	Chest pain	Myopericarditis	3	2	Schauer et al., 2021 [26]
BNT162b2, second dose	14	M	Chest pain, fever, headache	Myopericarditis	3	3	Schauer et al., 2021 [26]
BNT162b2, second dose	14	M	Chest pain, malaise, SOB	Myopericarditis	4	2	Schauer et al., 2021 [26]
BNT162b2, second dose	16	M	Chest pain, SOB	Myopericarditis	2	2	Schauer et al., 2021 [26]
BNT162b2, second dose	15	M	Chest pain	Myopericarditis	3	2	Schauer et al., 2021 [26]

M: male; F: female; SOB: shortness of breath.

## Data Availability

The datasets used during the current study are available from the corresponding author on reasonable request.

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
