# Peer review of "Safety, Immunogenicity, and Efficacy of COVID-19 Vaccines in Children and Adolescents: A Systematic Review"

_vaccines, 2021, doi:10.3390/vaccines9101102_

Round 1
Reviewer 1 Report
Lv and coworkers in their systematic review address an important issue of efficacy and safety of covid19 vaccine in children and adolescents. The systematic is important at this particular time where vaccine against Covid-19 is still far from the administration. The article is well within the scope of the journal however, the way a systematic review should be conducted should be improved. To least review this article in a better light I recommend authors to give a proper search strategy outline that can be validated.
Search strategy should be properly mentioned. It is often with AND OR Boolean system that results in a particular number of results. I am not able to validate the results using the search criteria mentioned below by the authors.
"COVID-19”, “SARS-Cov-2”, “2019-nCov(????)”, “Coronavirus disease 2019”, “adolescent”, “young”, “pediatrics”, “children”, “infant”, “newborn”, “neonates”, “youth”, and “vaccine” does not yield the results that are shown in figure 1. The search strategy is the basis of the systematic review. Even supplementary figures do not help. I suggest authors resubmit the manuscript with appropriate search criteria including inclusion and exclusion criteria.
Author Response
Dear reviewer,
On behalf of all co-authors, I would like to thank you for your constructive comments, which have helped us considerably improve the paper. Please see our point-by-point response (in red) to the comments (Please see the attachment). We have also revised the entire manuscript for language, grammar and style.
Yours sincerely,
Yaolong Chen

Reviewer 2 Report
The manuscript is interesting; however, I have some queries that authors need to incorporate and extensively revise their manuscript.
General comments: The manuscript should be edited and proofread by a native speaker or by someone with expertise in academic writing. Authors should avoid the use of too many personal pronouns such as “we”. Avoid short sentences such as “We conducted a systematic review.”
Abstract
An abstract should provide readers with highlights of the information discussed in the manuscript. Please revise the abstract (rephrase and rewrite poorly written sentences).
Background
It is not well written.
Results:
Table 1. Rearrange all the columns in following order:
Name of vaccine > Participants>Sample Size>Follow-up duration>Study design>Country>Funding> Reference.
Table 2. Cite the reference in the last column
Table 3. Rearrange in the following order:
Vaccination>Age>Gender>Symptoms>Diagnosis>Time of onset>Length of Hospitalization> Case
Besides, all the tables are numbered two times.
Check for grammatical errors throughout the section.
Section conclusion
The section needs to be a bit more elaborate and emphasize the importance of the study.
Author Response

(The authors gave the same response as above.)

Round 2
Reviewer 1 Report
The authors have meticulously revised the manuscript and have provided a detailed search strategy that will help readers how the data was derived and processed. The manuscript has improved significantly after the revision. The authors have incorporated all the suggested changes. This systemic review is the demand of the current time. The manuscript is well within the scope of the journal and may be accepted for publication after moderate typographical and grammatical corrections. I congratulate the authors for a great job!
Reviewer 2 Report
The authors have addressed my comments adequately.